# Territories of Nerve Endings of the Medial Plantar Nerve within the Abductor Hallucis Muscle: Clinical Implications for Potential Pain Management

**DOI:** 10.3390/diagnostics14161716

**Published:** 2024-08-07

**Authors:** You-Jin Choi, Timm Joachim Filler, Michael Wolf-Vollenbröker, Ji-Hyun Lee, Hyung-Jin Lee

**Affiliations:** 1Department of Anatomy, School of Medicine, Konkuk University, Chungju 27478, Republic of Korea; cyj7797@kku.ac.kr; 2Institute for Anatomy I, University Hospital Düsseldorf (UKD), Heinrich Heine University (HHU), 40225 Düsseldorf, Germany; timm.filler@hhu.de (T.J.F.); wolfvoll@hhu.de (M.W.-V.); 3Department of Anatomy and Acupoint, College of Korean Medicine, Gachon University, Seongnam 13120, Republic of Korea; 4Department of Anatomy, School of Medicine, CHA University, Seongnam 13448, Republic of Korea

**Keywords:** abductor hallucis muscle, neuroanatomy, chronic pain, motor nerve ending, botulinum neurotoxin

## Abstract

This study aimed to elucidate the intramuscular distribution pattern of the medial plantar nerve and determine its motor nerve ending territories within the abductor hallucis muscle using modified Sihler’s staining and external anatomical landmarks. The study included 40 specimens of the abductor hallucis muscle (13 men and seven women) from formalin-fixed (ten cadavers) and fresh cadavers (ten cadavers), with a mean age of 77.6 years. The entry point of the medial plantar nerve into the muscle was examined, followed by Sihler’s staining to analyze the intramuscular distribution pattern and motor nerve ending location within the abductor hallucis muscle. Ultrasound- and palpation-guided injections were performed to verify the applicability of motor nerve ending location-based injections. The areas with the highest concentrations of nerve entry points and nerve endings were identified in the central portion of the muscle. Ultrasound- and palpation-guided injections were safely positioned near the densest nerve ending region of the muscle. These detailed anatomical data and injection methods would be beneficial for proceeding with safe and effective injection treatments using various analgesic agents to alleviate abductor hallucis muscle-associated pain disorders.

## 1. Introduction

Lower extremity pain, which is often associated with conditions such as plantar heel pain, fasciitis, tarsal tunnel syndrome, and hallux valgus, significantly affects global health. This pain diminishes quality of life and results in a substantial socioeconomic burden due to escalated healthcare costs and productivity loss [1,2,3,4].

Botulinum neurotoxin (BoNT) has emerged as an effective therapeutic agent for pain management [1,4]. Initially known for its use in neuromuscular disorders, its scope has been expanded to address chronic pain. Recent studies have highlighted the analgesic and anti-inflammatory properties of BoNT, confirming its efficacy in combating chronic muscular and neuropathic pain [1,2,3,4]. BoNT induces muscle paralysis by inhibiting the release of the neurotransmitter acetylcholine at the neuromuscular junction, leading to muscle relaxation and pain relief. Current studies corroborate the use of BoNT in managing conditions such as hallux valgus, plantar fasciitis, and plantar heel pain syndrome [1,2,3,4].

The abductor hallucis muscle (ABHM) is a potential target for BoNT administration in the treatment of various foot-related symptoms [1,4]. Applying BoNT to the ABHM mitigates pain by decompressing the medial and lateral plantar nerves between the ABHM and quadratus plantae muscle and the chiasma plantare as well [4]. Simons et al. identified myofascial trigger points within the ABHM. These trigger points, located in the region of the medial hindfoot, give rise to referred pain that extends to the medial side of the foot [5]. Subsequently, numerous anatomical studies have noted the motor point of the ABHM [6,7], a region closely associated with myofascial trigger points, which has proven effective in the treatment of BoNT in patients with plantar fasciitis or plantar heel pain syndrome [1,4,8,9].

Despite these promising findings, a significant gap remains in the literature concerning the ideal injection site for managing ABHM-related pain [1,4,5,10]. Moreover, the elusive nature of the territories of the medial plantar nerve distribution pattern and the exact locations of the motor nerve endings within the ABHM can increase side effects within nearby muscle structures and reduce treatment efficacy. Therefore, this study aimed to elucidate the intramuscular medial plantar nerve distribution pattern and determine the motor nerve ending territories within the ABHM with external anatomical landmarks using modified Sihler’s staining. This research will help to identify potential pain-originating points and propose optimal injection sites to enhance the efficiency of injectables, thereby potentially broadening the range of therapeutic options.

## 2. Materials and Methods

This study included 40 sides of the ABHM specimens (13 men and 7 women) from formalin-fixed (10 cadavers) and fresh cadavers (10 cadavers), with a mean age of 77.6 years (range, 49–90 years). Ten formalin-fixed cadavers were used to analyze the entry point of the medial plantar nerve, the intramuscular nerve distribution pattern, and nerve endings within the ABHM. Additionally, 10 fresh cadavers were used for the injection study: 5 for the US-guided injections and 5 for palpation-guided injections (Appendix A). We applied the same methodology to all male and female specimens. This study was conducted under the principles of the Declaration of Helsinki. It was approved by the Institutional Review Board of the College of Medicine, The Catholic University of Korea (Approval No. MC22SISI0099). Ten cadavers were obtained from The Catholic University of Korea and the remaining 10 cadavers were provided by Konkuk University. All donors or authorized representatives provided written informed consent for the use of cadavers as well as for using the related materials in future research. All cadavers had intact feet, with no medical history of trauma, obvious inflammation, or birth defects in the foot region.

We hypothesized that intramuscular nerve endings would be concentrated in specific areas of the ABHM. We evaluated the following parameters: (1) nerve entry point locations, (2) nerve ending distribution patterns, and (3) muscle region-specific distribution density.

The ABHM was exposed by carefully dissecting the skin, subcutaneous tissue, crural and plantar fasciae, and the flexor retinaculum from the ankle and foot regions. The medial plantar nerve was dissected to examine the muscle entry points. After verification of the entry point locations, the ABHM specimens were harvested and Sihler’s staining was conducted on the entire ABHM with the distributed medial plantar nerve.

### 2.1. Modified Sihler’s Staining

Modified Sihler’s staining was performed to identify the intramuscular nerve distribution pattern and concentrated locations of the nerve endings of the medial plantar nerve within the ABHM. The detailed preparation and Sihler’s staining protocol were as follows.

The staining process included a series of seven distinct phases. Initially, the specimens were preserved in a 10% formalin solution for approximately 1 week. Following preservation, the specimens were washed with running water for a day and then subjected to a 3% potassium hydroxide solution for 2–3 weeks to soften. Next, the specimens were immersed in Sihler I solution for decalcification and bleaching, then stained with Sihler II solution to highlight the nerve and muscle fibers. Subsequent destaining involved reimmersion in the Sihler I solution. After destaining, the specimens were immersed in water for neutralization and then clarified using pure formamide to enhance tissue transparency for a detailed study.

### 2.2. Analysis of Nerve Entry Points, Intramuscular Nerve Distribution Patterns, and Nerve Ending Areas

The stained specimens were inspected using an LED light box that provided sufficient light to reveal the intramuscular pathways of nerve branches. The nerve entry points were dissected and analyzed before Sihler’s staining to maintain their original location. The ABHM was virtually sectioned into four equal parts, each representing a quarter (25%) of the total length, labeled A, B, C, and D. Within each of these areas, the nerve entry points and areas where the nerve endings were detected were marked, and the data are presented as percentages (in areas A, B, C, and D). These comprehensive analytical methods identified the location of the most common nerve entry points and areas with the highest concentration of nerve endings in the ABHM.

### 2.3. Verification of the Injection Methods Based on the Motor Nerve Ending Location Analysis

Real-time B-mode US was performed using HS50 (Samsung, Seoul, Republic of Korea) interfaced with a linear array transducer (LA3-14AD; 3–14 MHz). After localizing the surface landmarks of the medial malleolus, navicular tuberosity, the base of the proximal phalanx of the great toe, and the medial tuberosity of the calcaneus through palpation, they were marked on the cadaver’s skin with a permanent pen. The transducer was placed horizontally at the line passing vertically through the navicular tuberosity and scanned transversely along the ABHM muscle fiber direction. The ABHM was located deep in the skin, and there was very thin subcutaneous fat tissue between the skin and the ABHM in every cadaver’s foot at the reference line (e.g., the vertical line passing the navicular tuberosity). After finding the ABHM, the transversely located transducer was moved toward the origin of the muscle to examine the shape of the muscle, which gradually became thicker. The ABHM was confirmed by passive abduction and adduction of the first phalanx of the great toe, which moves the ABHM but not the surrounding muscles. This anatomical information and other detailed steps were employed to ensure the identification of the ABHM during the US. Injections were performed using a 25 mm 23 G needle after estimating the target depth using US. Using a real-time technique, the probe was placed close to the puncture site and the needle was advanced under direct US guidance. Injections were administered transversely using the in-plane method.

To compare the application of the injection methods with US-guided injection, we used the pinch method to target the AHBM, which we regarded as a palpation-guided injection. First, we identified the navicular tuberosity to locate the injection site, where the nerve endings were mostly found. Second, we identified the most inferior border of the ABHM by palpating the muscle and pinching it using the thumb and index finger. Finally, the needle was advanced toward the origin of the muscle within the ABHM along the direction of the muscle fibers (e.g., transverse injection).

Both injections were performed using 0.25 cc of blue dye mixed with BoNTs. For each cadaver, the same technique was used on both sides. We used two distinct injection methods across ten feet each. After the US- and palpation-guided injections, expert anatomists dissected each specimen to determine whether the dye was properly targeted to most of the motor nerve ending regions. We also evaluated the accuracy of injections and their complications (defined as needle passage through unintended structures, such as significant neurovascular structures).

## 3. Results

### 3.1. Location of the Medial Plantar Nerve Entry Points of the ABHM

Multiple entry points were observed, mostly in the deeper portion of the ABHM, close to the quadratus plantae muscle. The medial plantar nerve entered the ABHM at the inferior or inferolateral aspects of the muscle (Figure 1). Among the four equally divided areas, the nerve entry point was only observed in the C area, which was approximately at the center of the muscle belly (40/40 cases, 100%).

### 3.2. Distribution Pattern of the Medial Plantar Nerve and the Densest Area of the Nerve Endings within the ABHM

The medial plantar nerve, stained violet, was broadly distributed throughout the ABHM. The branches of the medial plantar nerves within the muscle proceeded towards the origin, center, and insertion parts of the ABHM and extended in a radiated pattern from the inferior to the superior area. A branch of a small diameter was innervated to the insertion area and had a long path that contained a tendinous portion (Figure 2). A small nerve branched from the main trunk of the medial plantar nerve and extended posteriorly or to the origin, before passing beneath the lower edge of the ABHM.

The nerve endings of the medial plantar nerve were observed at several locations within each muscle, predominantly in areas B and C (Figure 3). In 100% of cases, nerve endings were identified in areas B and C, approximately in the central region of the ABHM. Additionally, they were found in areas A and D in 75% and 60% of cases, respectively (Table 1).

### 3.3. Ultrasound (US)- and Palpation-Guided Injections into the ABHM Based on the Densest Nerve Ending Area and Surface Landmarks

A single injection site for both US- and palpation-guided injections was determined based on the intramuscular distribution of the ABHM and surface landmarks (Figure 4). The injection point was targeted at the middle part of the ABHM, which corresponded to the B area and was the proximal or posterior two-third portion of the ABHM. The dye injected into the target area was found in the middle of the ABHM. Each injected dye was spread horizontally along the muscle fibers. Therefore, the proposed injection sites and two distinct methods can be regarded as accurate locations and methods for reaching the cluster area of the ABHM nerve ending. We determined this area (injection site) to be easily identified using the navicular tuberosity, which is a prominent point on the medial side of the foot.

## 4. Discussion

The ABHM is located in the superficial layer of the foot and extends from the medial process of the calcaneal tuberosity to the base of the proximal phalanx of the great toe. In addition, it covers the origin of the plantar vessels and nerves along the medial border of the foot and forms the medial border of the resulting tarsal tunnel [11]. The ABHM is a potential pain site on the medial side of the foot and is associated with hallux valgus, plantar fasciitis, and plantar heel pain syndrome [1,2,3,4]. Various therapeutic approaches, including dry needling and injection of anesthetics or BoNT into the ABHM, are used clinically to reduce the pain associated with the aforementioned disorders [1,2,3,4,5].

The techniques used for dry needling or injecting anesthetics and BoNT vary according to the approach route and main target area. Manual palpation and US imaging are commonly employed to aid injection techniques aimed at the ABHM [1,3,4,5,12]. Certain clinicians perform dry needling into the ABHM by palpating the myofascial trigger point or by identifying the local twitch response within the muscle [2,13]. Philip et al. described advancing the needle toward the origin of the ABHM to alleviate plantar fasciitis and plantar heel pain syndrome [4,8]. Omar et al. used US-guided BoNT injection into the ABHM belly at the widest point in the cross-section of distal tarsal tunnel syndrome [1]. However, because of the lack of comprehensive knowledge regarding muscle morphology, nerve distribution, and the location of the nerve endings within the ABHM, any recommendations or injections for that area must be regarded as provisional rather than definitive.

Comprehensive anatomical information is crucial for ensuring treatment efficacy, and factors such as the nerve entry point into the ABHM play a significant role in establishing safe and effective muscle injections. The ABHM is innervated by the medial plantar nerve, which originates from the tibial nerve. One of the rare classical and detailed descriptions of the innervation of this muscle can be found in the work of the Berlin anatomists Frohse and Fränkel [14]. They localized the origin of the motor branch for the ABHM from the medial plantar nerve at about 40 mm distal to the anterior circumference of the tuber calcanei. This nerve branch then extends a further 10 mm distally below the fascia of the ABHM to ramify into three main branches. The first and last branches extend further into the depth of the muscle to supply the deeper portions. The middle branch, on the other hand, supplies the superficial parts of the muscle. In general, the authors classify the ramification as tree-like and very fine. From these deeply located fine nerve branches, which can form anastomoses, one fine branch runs distally to the insertion of the muscle and one proximally to the origin of the muscle. Asayeon et al. reported that the medial plantar nerve enters the ABHM inferior and posterior to the navicular tuberosity, which can be regarded as a similar location to near the superior portion or border of the ABHM [7]. In contrast, Wada et al. reported that the ABHM mostly showed the medial plantar nerve entry point within the posterolateral portion of the muscle [6]. The present study is in line with Wada’s study, in which branches from the medial plantar nerve innervated the ABHM from the inferior to superolateral aspect of the muscle. We did not observe a nerve branch entering the ABHM near the navicular tuberosity, indicating that the nerve branch entered the ABHM at the superior portion or border of the muscle. The different observations between the studies could be attributed to different dissection approaches. Furthermore, we are in line with Frohse and Fränkel’s observation of a long distal and short recurrent proximal nerve branch that supports the insertion and origin of the ABHM [14]. Wada et al. reported that the anatomically revealed nerve entry point of the ABHM could be regarded as an ABHM myofascial trigger point within the muscle because the medial hindfoot region is a common trigger point of the ABHM [5,15]. In addition, this nerve entry point is electrically stimulated; therefore, together with muscle morphology, the location of the medial plantar nerve entry point could help a physician perform accurate stimulation, resulting in a high treatment success rate.

Knowledge of the topographical distribution of motor endplates in human muscle is important both for neurophysiologists performing electromyography and for clinicians performing muscle biopsies [16]. Although motor endplates are the primary site of action of BoNT, their location in the ABHM is unknown. Because the effect of BoNT relies on its uptake at the presynaptic membrane of the motor endplate, the injection should be administered into the motor point, which is a motor endplate-rich territory [17,18]. However, previous studies have suggested a motor point where the motor nerve branch enters the muscle belly [6,7]. Identifying the accurate location of the motor nerve ending, known as the motor point, is crucial when placing an electrode on the ABHM [7]. In addition, there are reports that efficient muscle paralysis induced by BoNT is achieved following injection in the vicinity of the motor endplate [19] and with motor endplate-targeted BoNT injection, rather than non-targeted injections [18].

Many studies have reported that territories of the motor endplate are located in the middle of the muscle belly [16,17,20,21]. However, based on previous anatomical and histological studies, the locations of the motor endplates are different between muscles and are not always located in the middle of the muscle belly. The motor endplates within the sartorius, gracilis, rectus femoris, and psoas muscles are scattered, indicating widespread distribution of the motor endplates along the muscle, therefore resulting in multiple injections [20,21,22,23]. Consequently, it is essential to identify and provide precise territories for the motor endplate of a specific muscle. In this study, the ABHM showed dispersed motor endplates. The motor nerve endings within the ABHM were mainly found within areas B and C, where the middle and posterior portions of the muscle were located. This result partially corresponds with the findings of a previous study showing that the ABHM causes pain primarily in the medial aspect of the heel, with some extension to the posteromedial heel and along the medial longitudinal arch [24]. Although the exact myofascial trigger point could not be identified using cadaveric specimens, both past and present studies suggest that the middle and origin sites of the ABHM may serve as potential pain sites along the medial side of the foot. Further clinical studies confirming these pain sites and assessing the efficacy of analgesic injections will bolster the findings of this study.

Another practical application of knowing of the exact ramifications is the possibility of using the ABHM in the plastic surgical treatment of wound healing defects in the area of the medial foot [25]. The pedicled abductor hallucis flap (AH flap) is a good option for the local coverage of chronic wounds in the hindfoot area [26]. For this purpose, the ABHM is usually released distally at its insertion and turned around the proximal vascular bundle. The disadvantage here is often the donor site defect (distal insertion and medial foot arch), as the lack of function of the lifted muscle often causes problems. One way to minimize these defects is the modified procedure with a split of the muscle in the longitudinal direction, as has already been shown for the AH flap by Wang et al. Wang et al. (2019) report on a case where the proximal pedicled AH flap is split longitudinally so that only the raised superficial portion is turned over and used for wound coverage [26]. The deeper portion, on the other hand, remains in its original position. According to our results, in such cases, the flap modification also leaves the strong innervation of the deep portion intact, so that preserved (partial) function of the ABHM can be expected. According to Wong (2007) [27], the ABHM plays a major role in the functional integrity of the medial foot arch.

US has several advantages over other imaging methods. First, it is cheaper than MRI and CT and is not more harmful than other methods that emit radiofrequency. Some studies have suggested the use of US to inject lidocaine or other substances into the ABHM to prevent various types of damage [1,2]. US is widely used to prevent iatrogenic side effects and examine the anatomical structures underlying the skin [9,28,29]. Therefore, to verify and apply our study results, we performed US-guided injections into the ABHM, in addition to typical blind injections. We used surface landmarks to locate the entire muscle as well as the approximate location of the nerve endings within the muscle. Because the results showed that the nerve endings were mostly found within the central portion of the muscle, we selected one bony landmark that helped position the US transducer at the center of the muscle transversely, parallel to the ABHM fiber direction. The majority of the injections into or near the ABHM were performed transversely rather than locating the transducer in a vertical direction because of nervous and vascular structural damage, as well as the lesser pain. Therefore, we used both US- and palpation-guided injections in the transverse direction. The palpation-guided injection was performed by pinching the ABHM with the thumb and index finger at the navicular tuberosity. After the injection, the colored dye was located within area B, which corresponded to the central portion of the muscle. The success rate for both injections was 100%; however, we recommend using US-guided injections to ensure accurate injection depth and safe and effective treatment of the ABHM. Therefore, combining the results of this study, the location of nerve endings based on surface landmarks (e.g., navicular tuberosity) with palpation- and US-guided injections would be helpful for effective and safe pain management of the AHBM in the foot region.

Here, the intramuscular nerve distribution of the ABHM and the location of nerve endings were revealed using a reliable whole-mount nerve staining method. However, this study had a few limitations. Sihler’s staining offers a less detailed location of the nerve ending within the muscle than microscopic evaluation. However, this is a reliable staining method, as the terminal portion of the myelin sheath is close (a few µm) to the neuromuscular junction [30]. Sihler’s staining stains the end portion of the myelinated nerve fiber close to the motor endplates. Given that BoNT diffuses a few centimeters from the injection site, the terminal myelinated portion, considered the nerve ending in this study, can be considered an indicator of the neuromuscular junction. Second, the sample size assessed in this study was small and all specimens used were from South Korea. Hence, a larger sample size, which may be generalizable among various populations, is needed. Further studies addressing these shortcomings may provide a better understanding of the nerve endings in the ABHM. Third, this study used cadavers from older adults. As it is not feasible to determine active myofascial trigger points in cadavers, further studies focusing on the correlation between the location of the active trigger point in the patient and the potential anatomical trigger point will provide further insight into pain management. Fourth, we used specimens from intact feet without specific pathologies to ensure a standardized baseline for our observations and measurements. The morphology of muscle or nerve distribution in patients with foot-related medical histories, including surgery, inflammation, or birth defects, may differ from its original state. This variation could significantly affect the results and interpretation of our study.

Specifically, pathological conditions can alter anatomical structures, making it challenging to obtain clear ultrasound images to accurately interpret muscles, stain nerves, and analyze their distribution. Furthermore, the bony landmarks may be displaced or altered, complicating the placement and delivery of injections. In clinical practice, injections are often administered to patients with various foot pathologies. This represents a limitation of our study, as the results may not be directly applicable in such cases.

In conclusion, using Sihler’s staining, the central portion (e.g., area B) was found to be a potential anatomical trigger point in the ABHM. The navicular tuberosity may be considered a user-friendly anatomical landmark to locate potential pain sites within the ABHM. A single potential injection site may be located at the center of the muscle. Anatomical knowledge of the location of the potential pain site and nerve endings of the ABHM can be used to define injection sites to manage pain disorders in the ABHM.

## 5. Conclusions

The results of this study enhance our anatomical understanding of the relationship between the medial plantar nerve and the ABHM, as well as the distribution of intramuscular motor nerve endings within the muscle. The location of nerve endings can be reliably estimated based on surface landmarks such as the navicular tuberosity. Utilizing US- and palpation-guided injections would be helpful for ensuring effective and safe pain management of the ABHM in the foot region. This anatomical knowledge is crucial for understanding and managing pain associated with the medial plantar nerve and the ABHM in the foot.

## Figures and Tables

**Figure 1 diagnostics-14-01716-f001:**
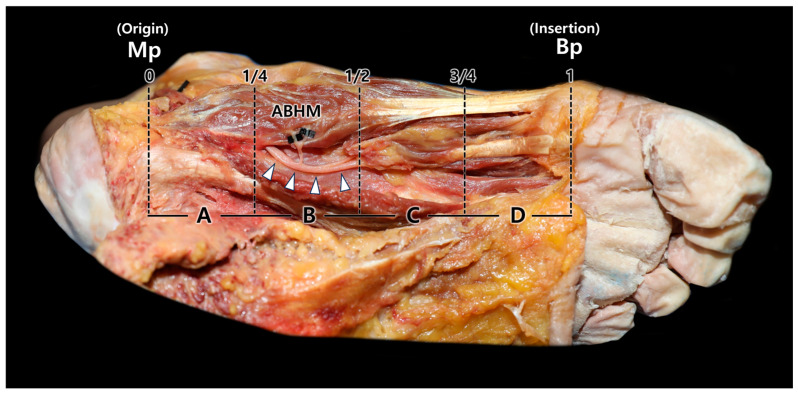
The inferior view of the abductor hallucis muscle (ABHM) and medial plantar nerve entry point into the ABHM. The image obtained from the cadaveric specimen shows the nerve entry point to the ABHM within four equally divided areas. White arrowheads indicate the medial plantar nerve and small black tapes were applied to visualize the nerve branches distributing to the ABHM. The medial plantar nerve mainly entered the ABHM at the inferior aspect of the muscle. Before piercing the muscle, the medial plantar nerve was divided into two or three branches distributed in a radiating pattern in area B. Mp, medial process of calcaneal tuberosity; Bp, the base of the proximal phalanx of the great toe.

**Figure 2 diagnostics-14-01716-f002:**
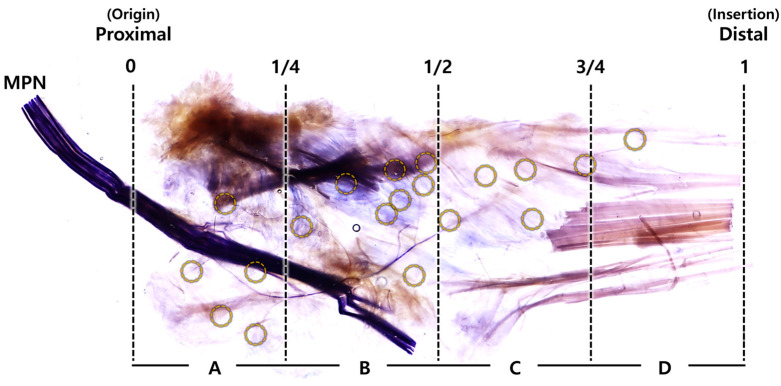
Photograph showing the intramuscular nerve distribution pattern of the medial plantar nerve (MPN) within the abductor hallucis muscle. The yellow circles indicate the nerve endings, with high density found in area B.

**Figure 3 diagnostics-14-01716-f003:**
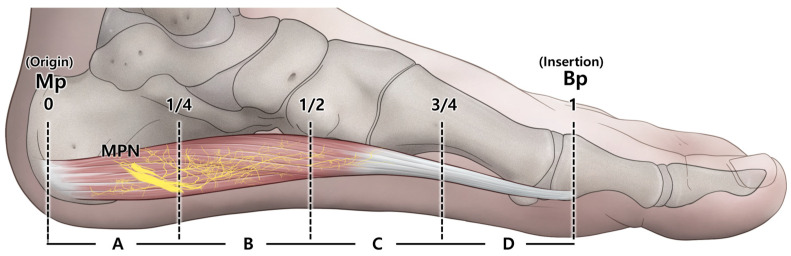
Illustration demonstrating the densest area of nerve endings of the MPN within the ABHM. The nerve endings of the MPN (medial plantar nerve) were mostly located in area B, corresponding to the region between the proximal one-fourth and the halfway point of the abductor hallucis muscle. The MPN trunk is deep within the abductor hallucis muscle and is indicated by the thick yellow band in the area A. Bp, base of the proximal phalanx of the great toe; Mp, medial process of the calcaneal tuberosity.

**Figure 4 diagnostics-14-01716-f004:**
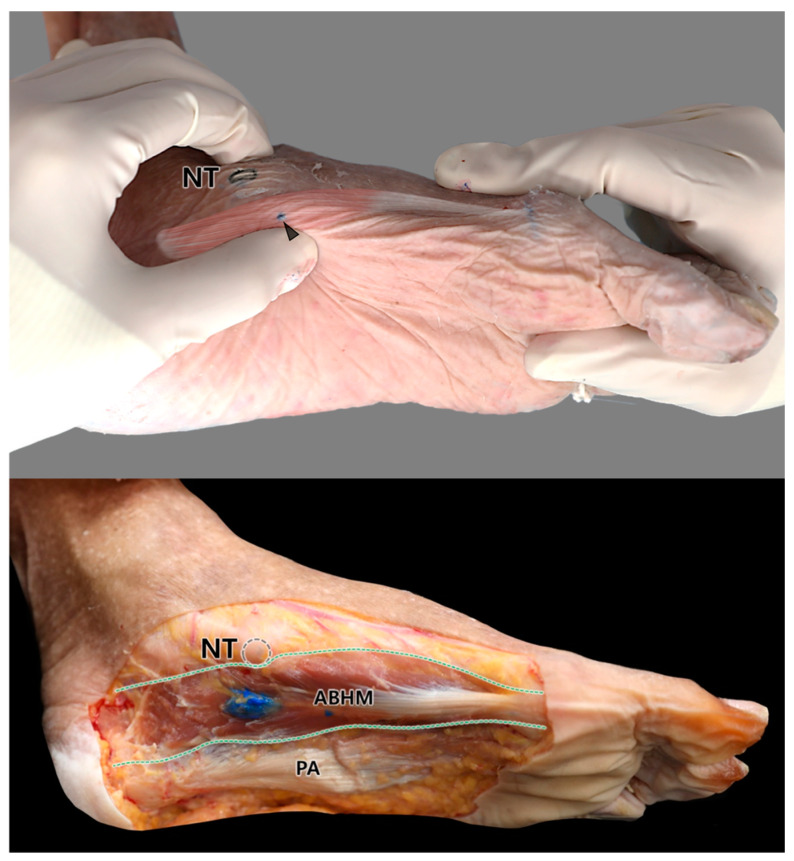
Cadaveric images show the surface landmark marked with a dashed black circle of the navicular tuberosity (NT) and the dissected cadaver’s foot following palpation-guided injection into the ABHM. The arrowhead in the upper image indicates the needle entry point and the green dashed lines in the bottom image indicate the upper and lower boundaries of the ABHM at the medial aspect of the foot. The blue dye is well positioned in area B, which corresponds to approximately one-fourth to one-half of the total length of the abductor hallucis muscle.

**Table 1 diagnostics-14-01716-t001:** Location of nerve endings of the medial plantar nerve within the ABHM at each section.

Specimens	Sides	Areas			
		A	B	C	D
1	Rt	X	O	O	X
	Lt	X	O	O	O
2	Rt	O	O	O	O
	Lt	X	O	O	O
3	Rt	X	O	O	O
	Lt	O	O	O	O
4	Rt	O	O	O	O
	Lt	X	O	O	O
5	Rt	X	O	O	O
	Lt	O	O	O	X
6	Rt	O	O	O	O
	Lt	O	O	O	O
7	Rt	O	O	O	O
	Lt	O	O	O	X
8	Rt	X	O	O	X
	Lt	O	O	O	O
9	Rt	O	O	O	O
	Lt	X	O	O	O
10	Rt	O	O	O	O
	Lt	O	O	O	X
Total		75%	100%	100%	60%

## Data Availability

The data presented in this study are available from the corresponding author upon reasonable request, as the cause of death and relevant pathology are considered personal data.

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
