# Peer review of "Territories of Nerve Endings of the Medial Plantar Nerve within the Abductor Hallucis Muscle: Clinical Implications for Potential Pain Management"

_diagnostics, 2024, doi:10.3390/diagnostics14161716_

Round 1
Reviewer 1 Report
Comments and Suggestions for Authors
In this study, the authors assessed the intramuscular distribution pattern of the medial plantar nerve in relation to the abductor hallucis muscle.
The study is interesting, however, several improvements need to be made.
Major comments
1) The authors need to provide more data on the cadaveric specimens used - usually in such research each specimen should be connected to an unique anonymized code, and at least the following data should be provided: sex, age at death, cause of death (if known), relevant pathology, methodology applied on specimen (US injection or palpation-guided, Sihler's staining, etc.). This can be provided as a supplementary table if it does not fit into the main text. From the current text, it is impossible to determine whether the formalin-fixed and fresh specimens were comparable, and whether the methodology was appropriately/equally applied in male and female specimens.
2) The authors need to better explain the study design in the Materials and Methods section. What is being evaluated (which parameters), how are these parameter being evaluated? Are there any comparisons being made? Is there a hypothesis being tested?
3) In the results section, the authors need to provide specific/concrete data for the research they conducted in either graphic or tabular format. A clarification in particular is necessary for lines 99 to 104. The authors claim that the axon terminals were evenly distributed throughout the ABMH - this needs to be substantiated by quantitative (numerical) data. The authors also claim that most axon terminal-concentrated areas were primarily identified in the C area - at first glance, this sounds a bit contradictory to the first statement (clearer wording is probably needed here), and, additionally, such a claim needs to be supported by quantitative data. Finally, the use of the percentages in the results section (e.g. 50 - 75%) is quite confusing - the authors seem to be referring to the position of their measurement, rather than the relative length. Perhaps using terms such as "proximal one fourth (1/4) of the muscle belly" would be more appropriate?
Minor comments
1) Lines 287-288: since the authors claim that the specimens used in this research had no pathology in the foot region, it should be discussed how this can affect the conclusions regarding the applications of injections. This might be especially relevant since one would presume that application of injection would be necessary when there is some pathology present. This does not invalidate the study, but it might be a limitation that needs addressing and mentioning in the discussion.
2) Line 290-291: "The medial and lateral plantar nerves were dissected to examine the entry points of the muscle." From the results it si not clear whether the dissection of the lateral plantar nerve was carried out, and if yes, how it is relevant to the study as presented.
Comments on the Quality of English LanguageThere are a few minor language issues:
line 41: "leading muscle relaxation" should probably be "leading to muscle relaxation"
lines 55-58: The logic of this sentence is strange. The first portion of the sentence states that the exact locations of axon terminals remain elusive, while the second portion states that this leads to reduced side effects and increase in treatment effect.
Author Response
Comments 1: The authors need to provide more data on the cadaveric specimens used - usually in such research each specimen should be connected to an unique anonymized code, and at least the following data should be provided: sex, age at death, cause of death (if known), relevant pathology, methodology applied on specimen (US injection or palpation-guided, Sihler's staining, etc.). This can be provided as a supplementary table if it does not fit into the main text. From the current text, it is impossible to determine whether the formalin-fixed and fresh specimens were comparable, and whether the methodology was appropriately/equally applied in male and female specimens.
Response 1: Thank you for your meaningful review and pointing this out. We agree with this comment. Therefore, we have provided several information as follows:
- Cadaver information: We have already provided the cadavers' sex and age at death in the Materials and Methods section. However, in our country, the cause of death and relevant pathology are considered personal data, so obtaining this information is impossible. Please understand our specific situation.
- Purpose of the specimen use: The methodology applied to the specimens (US injection or palpation-guided, Sihler's staining, etc.) is detailed in the Materials and Methods section. We have included detailed information regarding the use of formalin-fixed and fresh specimens in the study as follows, on page 7, lines 289-293:
“Ten formalin-fixed cadavers were used to analyze the entry point of the medial plantar nerve, intramuscular nerve distribution pattern, and nerve endings within the ABHM. Additionally, ten fresh cadavers were used for the injection study: five for the US-guided injections and five for palpation-guided injections (Suppl. Table 1). We applied the same methodology to all male and female specimens.”
Based on the above information, we did not compare any anatomical structures between formalin-fixed and fresh specimens because different cadavers were used for different purposes.
- Methodology application: We have applied the same methodology for all male and female specimens. Therefore, we have included the following sentence in the Materials and Methods section, on page 7, line 293:
" We applied the same methodology to all male and female specimens."
Comments 2: The authors need to better explain the study design in the Materials and Methods section. What is being evaluated (which parameters), how are these parameter being evaluated? Are there any comparisons being made? Is there a hypothesis being tested?
Response 2: Thank you for your meaningful review and for pointing this out. We have added the following sentence to improve the understanding of our analysis methods for nerve entry points, most-concentrated nerve ending areas, and hypothesis, on page 8, lines 302-304, and lines 332-333:
“We hypothesized that intramuscular nerve endings would be concentrated in specific areas of the ABHM. We evaluated the following parameters: 1) nerve entry point locations, 2) nerve ending distribution patterns, and 3) muscle region-specific distribution density.”
“Within each of these areas, the nerve entry points and areas where the nerve endings were detected were marked; the data are presented as percentages (in areas A, B, C, and D)”
Comments 3: In the results section, the authors need to provide specific/concrete data for the research they conducted in either graphic or tabular format. A clarification in particular is necessary for lines 99 to 104. The authors claim that the axon terminals were evenly distributed throughout the ABMH - this needs to be substantiated by quantitative (numerical) data. The authors also claim that most axon terminal-concentrated areas were primarily identified in the C area - at first glance, this sounds a bit contradictory to the first statement (clearer wording is probably needed here), and, additionally, such a claim needs to be supported by quantitative data. Finally, the use of the percentages in the results section (e.g. 50 - 75%) is quite confusing - the authors seem to be referring to the position of their measurement, rather than the relative length. Perhaps using terms such as "proximal one fourth (1/4) of the muscle belly" would be more appropriate?
Response 3: Thank you for your valuable insight into our results. According to the reviewer’s suggestion, we have provided quantitative (numerical) data regarding the location of the axonal terminals as shown in Table 1 in the Results section. Following the reviewer’s advice, we have chosen not to use percentages in the Results. Since our data showed that the nerve endings were mainly observed within the central portion of the ABHM, we did not use the term "proximal one-fourth" as suggested by the reviewer. Instead, we have focused on providing the most concentrated area of the nerve endings, so we have referred to the "central portion of the ABHM." Therefore, we have made changes to the Results according to the reviewer’s suggestions as follows, on page 3, lines 93-97:
“The nerve endings of the medial plantar nerve were observed at several locations within each muscle, predominantly in areas B and C (Figure 3). In 100% of cases, nerve endings were identified in areas B and C, approximately in the central region of the ABHM. Additionally, they were found in areas A and D in 75% and 60% of cases, respectively (Table 1).”
In addition to the above change, we have changed Figure 2 and its figure legend (on page 3、 lines 89-92)and modified Figure 3 and its figure legends (on page 3、 lines 93-104)to better show the data as follows:
Table 1. Location of axonal terminals of the medial plantar nerve within the ABHM at each section
|
Specimens |
Sides |
Areas |
|||
|
A |
B |
C |
D |
||
|
1 |
Rt |
X |
O |
O |
X |
|
Lt |
X |
O |
O |
O |
|
|
2 |
Rt |
O |
O |
O |
O |
|
Lt |
X |
O |
O |
O |
|
|
3 |
Rt |
X |
O |
O |
O |
|
Lt |
O |
O |
O |
O |
|
|
4 |
Rt |
O |
O |
O |
O |
|
Lt |
X |
O |
O |
O |
|
|
5 |
Rt |
X |
O |
O |
O |
|
Lt |
O |
O |
O |
X |
|
|
6 |
Rt |
O |
O |
O |
O |
|
Lt |
O |
O |
O |
O |
|
|
7 |
Rt |
O |
O |
O |
O |
|
Lt |
O |
O |
O |
X |
|
|
8 |
Rt |
X |
O |
O |
X |
|
Lt |
O |
O |
O |
O |
|
|
9 |
Rt |
O |
O |
O |
O |
|
Lt |
X |
O |
O |
O |
|
|
10 |
Rt |
O |
O |
O |
O |
|
Lt |
O |
O |
O |
X |
|
|
Total |
|
75% |
100% |
100% |
60% |
Comments 4: Lines 287-288: since the authors claim that the specimens used in this research had no pathology in the foot region, it should be discussed how this can affect the conclusions regarding the applications of injections. This might be especially relevant since one would presume that application of injection would be necessary when there is some pathology present. This does not invalidate the study, but it might be a limitation that needs addressing and mentioning in the discussion.
Response 4: Thank you for your meaningful review and for pointing this out. We agree with your comment and have added information as a limitation of the study in the discussion section about how the presence of pathology in the foot region could affect the application of injections. We highlighted the added sentence in red color below.
Before presenting the specific changes we made, we would like to share a brief opinion to help clarify our concept. Innervation is a strong and relatively constant characteristic of muscles. Except for accompanying demyelinating nerve disorders or other disorders of peripheral nerves (like diabetes or Guillain-Barré syndrome), it is not expected that the general morphology of nerves will be altered in cases when a botulinum toxin injection is needed. The presence of pathology in the foot region may indeed affect the outcomes of injection therapies. However, as the first step in developing novel therapies, it is crucial to understand the conditions in healthy subjects. Our study focuses on examining the course of the ABHM's nerve supply in healthy cadavers to establish a baseline.
We have added the following paragraphs on page 7, lines 260-270.
"Fourth, we used specimens from intact feet without specific pathologies to ensure a standardized baseline for our observations and measurements. The morphology of muscle or nerve distribution in patients with foot-related medical histories, including surgery, inflammation, or birth defects, may differ from its original state. This variation could significantly affect the results and interpretation of our study.
Specifically, pathological conditions can alter anatomical structures, making it challenging to obtain clear ultrasound images to accurately interpret muscles, stain nerves, and analyze their distribution. Furthermore, the bony landmarks may be displaced or altered, complicating the placement and delivery of injections. In clinical practice, injections are often administered to patients with various foot pathologies. This represents a limitation of our study, as the results may not be directly applicable in such cases.”
Comments 5: Line 290-291: "The medial and lateral plantar nerves were dissected to examine the entry points of the muscle." From the results it si not clear whether the dissection of the lateral plantar nerve was carried out, and if yes, how it is relevant to the study as presented.
Response 5: Thank you for your meaningful review and for pointing this out. We agree with this comment. Therefore, we have made changes to the sentence as follows, on page 8, lines 306-307:
“The medial plantar nerve was dissected to examine the muscle entry points."
Comments 6: There are a few minor language issues:
line 41: "leading muscle relaxation" should probably be "leading to muscle relaxation"
lines 55-58: The logic of this sentence is strange. The first portion of the sentence states that the exact locations of axon terminals remain elusive, while the second portion states that this leads to reduced side effects and increase in treatment effect
Response 6: Thank you for your meaningful review and pointing this out. We agree with this comment. Therefore, we have made the changes as per the reviewer’s suggestion.
On page 1, lines 41-42: "leading muscle relaxation" has been corrected to "leading to muscle relaxation"
on page 2, lines 55-58: We have revised the text as follows, based on the reviewer’s suggestion: “Moreover, the elusive nature of the territories of the medial plantar nerve distribution pattern and the exact locations of the motor axon terminals within the ABHM can lead to increased side effects on nearby muscle structures and a reduction in treatment efficacy.”
Reviewer 2 Report
Comments and Suggestions for Authors
I reviewed the excellent article titled - The Intramuscular Distribution of the Medial Plantar Nerve and Motor Axon Terminals within the Abductor Hallucis Muscle: Implications for Pain Management. The manuscript by Choi et al., presents a detailed anatomical study focusing on the medial plantar nerve's entry points and the distribution of motor axon terminals within the abductor hallucis muscle (ABHM). The authors have employed a modified Sihler's staining to elucidate these anatomical details, proposing optimal injection sites for pain management interventions. I liked the paper specially because the findings have possible clinical applications, especially in improving the effectiveness of Botulinum neurotoxin (BoNT) injections for addressing foot pain disorders.
If revising, please consider including a brief section on potential clinical applications in the conclusion or discussion section to emphasize the practical relevance of the findings. The elucidation of possible problems/side effects can also be improved.
The quality of English is good with minor editing requested.
Author Response
Comments 1: If revising, please consider including a brief section on potential clinical applications in the conclusion or discussion section to emphasize the practical relevance of the findings. The elucidation of possible problems/side effects can also be improved.
Response 1: We have already provided potential clinical applications for surgical treatment on page 6, lines 205-220. Given that this is a cadaveric study and should be evaluated in the clinical field to immediately apply the results and injection techniques, we have verified our injection techniques using ultrasonography- and palpation-guided injections. These techniques can be helpful for those who perform botulinum neurotoxin injections into the abductor hallucis muscle. In addition to the above-mentioned clinical application, we have included the following description in Conclusions, on page 7, lines 280-283, to facilitate the application of our results more easily in the clinical field.
“The location of nerve endings can be reliably estimated based on surface landmarks such as the navicular tuberosity. Utilizing US- and palpation-guided injections would be helpful for ensuring effective and safe pain management of the ABHM in the foot region.”
Comments 2: Comments on the Quality of English Language. The quality of English is good with minor editing requested.
Response 2: Thank you for your meaningful review and for pointing this out. We agree with this comment. Therefore, we have had the entire manuscript re-evaluated by an English editing service.
Reviewer 3 Report
Comments and Suggestions for Authors
In general, as a neuroanatomist specializing in Neural Engineering, I find the paper interesting, scientifically relevant, and well written. Only minor adjustments are necessary.
Page 1, lines 22-23: "The areas with the highest concentrations of nerve entry points and axon terminals were identified in the central portion of the muscle." This statement is qualitative. To improve, I suggest including in the "next steps" section (or in the end of conclusion) that image processing to quantify the percentage concentration of nerve entry points may be applied in future studies.
Keywords: To enhance discoverability in databases following the paper's publication, refrain from duplicating terms from the title, such as "Abductor hallucis muscle; medial plantar nerve; pain." Opt for synonymous terms to broaden search relevance, such as "Neuroanatomy, Spinal Nerve," or others.
Figure 3: Was the illustration adapted from another image? If yes, please indicate the source.
Figure 4: The abbreviation "NT" was not defined.
Author Response
Comments 1: Page 1, lines 22-23: "The areas with the highest concentrations of nerve entry points and axon terminals were identified in the central portion of the muscle." This statement is qualitative. To improve, I suggest including in the "next steps" section (or in the end of conclusion) that image processing to quantify the percentage concentration of nerve entry points may be applied in future studies.
Response 1: Thank you for your meaningful review and for pointing this out. Due to the abstract word limit, we have described the most concentrated area for nerve entry points and nerve endings as the central portion of the muscle, rather than using a quantitative value. We believe that the description of the central portion of the muscle makes it easier for clinicians to understand the location of the nerve entry points and nerve endings, and to apply the results in the clinical field. To improve the results, we have provided detailed quantitative data in the Results, specifically in Table 1. In particular, we have only provided the quantitative data for the location of the motor nerve endings because the nerve entry points were found in the central portion of the muscle in all cases, as described in the Results section.
Table 1. Location of axonal terminals of the medial plantar nerve within the ABHM at each section
|
Specimens |
Sides |
Areas |
|||
|
A |
B |
C |
D |
||
|
1 |
Rt |
X |
O |
O |
X |
|
Lt |
X |
O |
O |
O |
|
|
2 |
Rt |
O |
O |
O |
O |
|
Lt |
X |
O |
O |
O |
|
|
3 |
Rt |
X |
O |
O |
O |
|
Lt |
O |
O |
O |
O |
|
|
4 |
Rt |
O |
O |
O |
O |
|
Lt |
X |
O |
O |
O |
|
|
5 |
Rt |
X |
O |
O |
O |
|
Lt |
O |
O |
O |
X |
|
|
6 |
Rt |
O |
O |
O |
O |
|
Lt |
O |
O |
O |
O |
|
|
7 |
Rt |
O |
O |
O |
O |
|
Lt |
O |
O |
O |
X |
|
|
8 |
Rt |
X |
O |
O |
X |
|
Lt |
O |
O |
O |
O |
|
|
9 |
Rt |
O |
O |
O |
O |
|
Lt |
X |
O |
O |
O |
|
|
10 |
Rt |
O |
O |
O |
O |
|
Lt |
O |
O |
O |
X |
|
|
Total |
|
75% |
100% |
100% |
60% |
Comments 2: Keywords: To enhance discoverability in databases following the paper's publication, refrain from duplicating terms from the title, such as "Abductor hallucis muscle; medial plantar nerve; pain." Opt for synonymous terms to broaden search relevance, such as "Neuroanatomy, Spinal Nerve," or others.
Response 2: Thank you for your meaningful review and for pointing this out. We agree with this comment. Therefore, we have changed the keywords as follows on page 1, lines 28-29:
“Keywords: neuroanatomy, chronic pain“
Comments 3: Figure 3: Was the illustration adapted from another image? If yes, please indicate the source.
Response 3: Thank you for your meaningful review and for pointing this out. We created the illustration ourselves, therefore, we did not indicate the source of the figure.
Comments 4: Figure 4: The abbreviation "NT" was not defined.
Response 4: Thank you for your meaningful review and for pointing this out. We have defined “NT” as “navicular tuberosity” in the figure legend of Figure 4 as follows, on page 4, line 119:
“Figure 4. Cadaveric images show the surface landmark marked with a dashed black circle of the navicular tuberosity (NT) and ~~”
Round 2
Reviewer 1 Report
Comments and Suggestions for Authors
The authors adequately addressed my comments.
I have a minor suggestion - perhaps it would be beneficial for readers to state (maybe in the supplement?) that some information is private data, which could not be provided, even under code (I'm refering to this statement: "the cause of death and relevant pathology are considered personal data, so obtaining this information is impossible").